# A comparative study of hematological parameters between hypertensive and normotensive individuals in Harar, eastern Ethiopia

Beza Sileshi[1], Fekadu Urgessa[2], Moges Wordofa[2]*

1 Department of Medical Laboratory Science, Collage of Health and Medical Sciences, Haramaya University, Harar, Ethiopia, 2 Department of Medical Laboratory Science, Collage of Health and Medical Sciences, Addis Ababa University, Addis Ababa, Ethiopia

* heranmakmow@gmail.com

## Abstract

### Background

Hypertension is the major public health concern; leading to cardiovascular disease. It is associated with alteration in hematological parameters which may lead to end-organ damage. Thus, this study aimed to compare hematological parameters between hypertensive and normotensive adult groups in Harar, eastern Ethiopia.

### Methods

A comparative cross-sectional study was conducted from January to March, 2020 at Jugel and Hiwotfana Specialized University hospital, Harar, eastern Ethiopia. Convenient sampling technique was used to recruit 102 hypertensive patients from the two hospitals and 102 apparently healthy blood donors. Participant's socio-demographic and clinical information were collected using pre-tested structured questionnaire. Blood sample were collected and analyzed by Beckman Coulter DxH 500 analyzer for complete blood count. The data were entered and analyzed using SPSS version 23. Independent **t**-test and Mann Whitney u-test was used for comparison between groups. Spearman's correlation was used for correlation test. P values less than 0.05 was considered as statistically significant.

### Result

102 hypertensive and 102 healthy controls were enrolled in this study. The median ± IQR value of white blood cell (WBC) count, hemoglobin (Hgb), hematocrit (HCT), red cell distribution width (RDW) and mean platelet volume (MPV) were significantly higher in hypertensive group compared to apparently healthy control group. Additionally, RBC (red blood cell) count, HCT and RDW showed statistically significant positive correlations with systolic and diastolic blood pressure. WBC count and RDW were significantly and positively correlated with body mass index (BMI). Platelet (PLT) count had a significant but negative correlation (r

**Data Availability Statement:** All the relevant data are within the manuscript and its supporting information file.

**Funding:** This work was financially supported by Addis Ababa University, Ethiopia. The funder has no role in the study selection, data collection, analysis, conclusion, and interpretation.

**Competing interests:** The authors have declared no competing interest.

= -0.219, P = 0.027) with duration of hypertension illness while MPV showed positive and significant correlation (r = 0.255, P = 0.010).

## Conclusion

The median values of WBC, Hgb, HCT, RDW and MPV were significantly higher in hypertensive patient compared to apparently healthy individuals. Hence, it is important to assess hematological parameters for hypertensive individuals which may help to prevent complications associated with hematological aberrations. However, further studies are required to understand hypertensive associated changes in hematological parameters.

## Introduction

Hypertension (HTN) is defined as high blood pressure. Blood pressure is the power of blood exerted to the walls of arteries as the heart pumps blood [1]. Normal range of blood pressure is defined as, <120mmHg for systolic blood pressure (SBP) and <80 mmHg for diastolic blood pressure (DBP). Hypertension is a reading of SBP ≥140 mmHg and/or DBP ≥90 mmHg [2]. The pathophysiology of hypertension is unknown. In about 95–97% of hypertension cases, the cause is unknown and its characterized as primary hypertension. In the remaining small percent, an underlying disease is responsible for the raised blood pressure [3].

Often times, hypertension shows no symptoms until it has done substantial damage to the cardiovascular system. Thus, its often referred as "silent killer" [4]. A high blood pressure could result blood vessels to develop bulges and create weak spots; which makes it to collapse and burst. If left uncontrolled, hypertension could cause a heart attack, kidney failure, stroke and sometimes death [5].

The cellular components of blood contribute to the viscosity, volume and coagulability of the blood; thus, playing a vital role in regulating blood pressure [6]. Hypertension can alter hematological parameters of the body, leading to functional disturbances in many systems of the body. It causes an increase in WBC count, a decrease in red cell deformability and an increase in platelet activation [7–9]. This alteration may worsen in the microcirculation and enhance an end-organ damage [10].

Hypertension is the leading cause of cardiovascular disease (CVD) and premature death worldwide. 1.13 billion people are affected by HTN worldwide [11]. Though, hypertension is a preventable and modifiable risk factor of CVD, its prevention and control has not yet received appropriate attention in many developing countries [12]. Ethiopia is one of the lower income countries that is troubled by double burden of the diseases. A surveillance in Addis Ababa reported that, 51% of all deaths were due to non-communicable disease, of which, CVD accounts for 24% and hypertension was responsible for 12% of the CVD deaths [13].

Many recent guidelines on the diagnosis and management of hypertension emphasis that, total risk of cardiovascular disease should be quantified so that the type and intensity of treatment can be personalized to the degree of overall risk rather than the level of blood pressure (BP) elevation alone [14]. So, identifying patients with high risk of developing cardiovascular disease (CVD) will enable for early administration of hypertensive treatment and thus, reduce the progression of silent vascular damage [15].

There has been a detailed study of the relationship between high blood pressure and biochemical risk factors, like lipid profile association with hypertension and CVD [6]. However, the definitive pathophysiologic mechanism of hypertension involving hematological parameters is not clearly understood.

While some research has been carried out to investigate the relationship between hematological parameters and hypertension, there is still a broad variation in hematological profile of hypertensive patients between different researches worldwide. Moreover, there are only a few studies conducted on hematological parameters of hypertensive patients in Ethiopia. Studies have shown hypertension related changes in hematological parameters and investigation of these parameters might help for earlier identification and timely management and monitoring of hypertension related complication such as cardiovascular disease. Thus, routinely available and inexpensive hematological test such as complete blood count (CBC) will enable the physician for periodic assessment of the patient conditions and early intervention. So, this study aimed to compare hematological parameters between hypertensive patients and apparently healthy normotensive individuals.

## Material and methods

### Study design, setting and population

A comparative cross-sectional study design was conducted from January to March, 2020 in two public hospitals; Hiwotfana Specialized University hospital (HFSUH) and Jugel hospital, Harar, Eastern Ethiopia. HFSUH is located in Harar town, 527 kilometer east of Addis Ababa, the capital city of Ethiopia [16]. The hospital is a tertiary referral hospital affiliated with the College of Health and Medical Sciences of Haramaya University, Ethiopia. It is the major referral hospital in the eastern part of the country serving a catchment area with a population close to 3 million [17]. Jugel Hospital is a regional general hospital found in the same town and run by the Harari Regional Health Bureau.

A total of 204 (102 confirmed hypertensive patients and 102 apparently healthy blood donors) study participants were recruited using convenient sampling method, meaning that after reviewing their medical records as well as in depth face to-face interview with the participant, only who fulfilled the eligibility criteria and consented to participate were selected. There was a total of 80 and 70 hypertensive patients on follow up in HFSUH and Jugel hospitals, respectively. Thus, we used proportional allocation method to recruit participants from the two hospitals. Accordingly, 54 and 48 patients were selected from HFSUH and Jugel hospital, respectively. A study participant with hypertensive and healthy control group were age and sex matched. At Harar blood bank, donors who were screened for transfusion transmitted infectious disease and other chronic diseases and found their blood to be safe for transfusion and gave their consent were recruited and considered as control (normotensive) group. Hypertensive patients having a history of infectious and chronic disease, alcohol consumers, smokers, patients taking antibiotic, treatment for anemia, patients with systemic diseases, pregnant woman and those with secondary hypertension were excluded.

### Data collection procedure

Before the data collection, written informed consent was obtained from each participant. In-depth interview using structured questionnaire and review of medical records was used to obtain socio-demographic and clinical data of participant. Prior to the actual data collection, the questionnaire which designed in English was translated into local language (Afan Oromo and Amharic) to obtain more reliable data. The questionnaire was pretested among non-participants at the study site to test its validity. A 3 ml of venous blood was collected from participants by qualified phlebotomist and the sample was run in hospital laboratory by qualified laboratory professional. Blood pressure measurement and anthropometric data such as height and weight were also collected from each participant.

## Blood pressure and anthropometric measurement

The data regarding anthropometric variables such as height (to the nearest centimeter without shoes) and weight (to the nearest 0.1 kg) was collected and BMI was calculated as weight in kilograms divided by height in meter squared. Blood pressure was measured by qualified personnel (nurse) using analog sphygmomanometer and stethoscope. For blood pressure determination, measurement was taken twice and average value was used.

## Laboratory analysis

A 3 ml venous blood was collected from each participant by experienced phlebotomist and the sample was analyzed immediately by Beckman Coulter DxH 500 hematology analyzer for complete blood count. WBC count, red blood cell parameters (Hgb, HCT, MCV, MCH, MCHC, RDW) and platelet parameters (PLT count, MPV, PDW) were obtained from each sample.

## Data quality assurance

The sample was collected and analyzed by qualified laboratory professional by strictly following standard operation procedures. In addition, three level (low, normal and high) commercially prepared control materials were used daily. Completeness and reliability of the data was properly checked and result were reviewed and properly documented.

## Operational definition

**Hypertension.**   Hypertension is defined as systolic blood pressure $\geq 140$ mmHg or diastolic blood pressure $\geq 90$ mmHg.

**Pre hypertension.**   Is defined as systolic blood pressure of 120 -139mmHg and or diastolic pressure of 80–89 mmHg.

**Stage 1 hypertension.**   A sub type of hypertension with systolic pressure of 140-159mmHg or diastolic pressure of 90–99 mmHg.

**Stage 2 hypertension.**   Is a subtype of hypertension with systolic pressure of greater than or equal to 160mmHg or diastolic pressure of greater than equal to 100 mmHg [2].

**Case.**   Patient who are confirmed to be hypertensive and on follow up at Jugel and Hiwotfana Specialized University hospital and fulfill the inclusion criteria.

**Control.**   Normotensive apparently healthy screened blood donors at Harar blood bank and fulfill the inclusion criteria.

## Statistical analysis

Data was entered and analyzed using SPSS version 23 software. The results are presented using tables. The normality of data distribution was checked by statistical tools of Kolmogorov-Smirnov (K-S). Mean±SD was used for normally distributed data while median with interquartile range for non-parametric data. Comparison of hematological parameters between hypertensive patient and apparently healthy normotensive controls was done with independent t-test for normally distributed data and Mann-Whitney U test for non-parametric data. The correlation of hematological parameters with blood pressure indices (systolic blood pressure, diastolic blood pressure) was assessed by Spearman's correlation. P value $< 0.05$ was considered as statistically significant.

## Ethical consideration

Ethical clearance was obtained from the department of Research and Ethics Review Committee (DRERC) of Addis Ababa University, College of Health Sciences, department of Medical Laboratory Science and it was in accordance with the principles of the Helsinki II declaration.

In addition, an official letter of permission was obtained from the hospitals. Written informed consent was obtained from each participant and confidentiality of the data was assured throughout the process.

## Results

### Sociodemographic and clinical characteristics of the study participants

The study included 204 study participants that comprise, 102 hypertensive patients with mean age of 51.11 ± 8.4 years and 102 controls with mean age of 48.65 ± 8.85 years. 63 (61.8%) of participants were females in both hypertensive and healthy control groups. The majority of hypertensive patients (85.3%) and control group (83.3%) were urban dwellers. Most of the participants in hypertensive patients, 60 (58.8%) and healthy control group, 48 (47.1%) were married. Regarding educational status, 44 (43.1%) of hypertensive had primary school while 59 (57.9%) of control had up to secondary school.

The mean value of the systolic and diastolic pressure of hypertensive patients were 138.36 ±16.25 and 85.61 ±9.28, respectively. The mean body mass index of the patients was 24.22 ± 3.93 while it is 23.37 ± 3.57 for control groups. The stage of blood pressure of the patients was also determined and 8.8%, 38.2%, 37.3% and 15.7% were staged as normal, prehypertension, stage 1 and stage 2 respectively (Table 1).

### Comparison of hematological parameters among hypertensive and control groups

The normality of the variables was tested and all hematological parameters except platelets were not normally distributed. As result, all variables except platelets are expressed as median plus interquartile range and compared by Manny Whitney test while platelet count is expressed by mean ± SD and tested by independent t test.

The result showed that, there was a statistically significant increase in WBC count, HGB, HCT, RDW and MPV in hypertensive compared to a healthy control group. In contrast, RBC, MCV, MCHC, MCH and PLT showed no statistically significant difference between the two groups (Table 2).

### Correlation of hematological parameters with blood pressure indices among hypertensive individuals

A spearman's correlation was used to assess the relationship between various hematological parameters and the blood pressure indices among hypertensive individuals. The result showed that, there was a significant and weak positive correlation between RBC count and SBP (r = 0.255, P = 0.01) as well as RBC and DBP (r = 0.241, P = 0.015). HCT and RDW also revealed a significant positive correlation with systolic and diastolic blood pressures. The other hematological variables like WBC, MCV, MCHC, MCH, RDW-SD, PLT count, and MPV didn't show significant correlation with blood pressure indices (Table 3).

### Correlation of hematological parameters with duration of illness and body mass index

Spearman correlation was also done to assess linear relationship between some hematological parameters and body mass index. RDW and WBC count demonstrated a significant but weak positive correlation with BMI. On the other hand, MPV achieved significant positive correlation with duration of illness while PLT count had negative correlation (Table 4).

**Table 1. Socio-demographic and clinical characteristics of study participants at HFSUH and Jugel Hospitals, Harar, eastern Ethiopia, 2020 (n = 204).**

| Variables | | Hypertensive (n = 102) | Healthy Control (n = 102) |
|---|---|---|---|
| Age | 18–35 | 4(3.9%) | 6(4.9%) |
| | 36–55 | 63(61.8%) | 60(59.8%) |
| | >55 | 35(34.3%) | 36(33.3%) |
| | Mean ±SD | 48.85 ± 8.99 | 51.1 ± 8.4 |
| Sex | Male | 39(38.2%) | 39(38.2%) |
| | Female | 63(61.8%) | 63(61.8%) |
| Residence | Urban | 87(85.3%) | 85(83.3%) |
| | Rural | 15(14.7%) | 17(16.7%) |
| Marital status | Single | 7(6.9%) | 34(33.3%) |
| | Married | 60(58.8%) | 48(47.1%) |
| | Divorced and widowed | 35(35.3%) | 20(19.6%) |
| Educational status | Illiterate[d] | 22(21.6%) | 12(11.8%) |
| | Primary (1–8) | 44(43.1%) | 16(15.7%) |
| | Secondary (9–12) | 28(27.4%) | 59(57.9%) |
| | Diploma and above | 8(7.8%) | 15(14.7%) |
| Body mass index | Mean ± SD | 24.22 ± 3.93 | 23.37 ± 3.57 |
| Blood Pressure Indices | Systolic BP (Mean ± SD) | 138.36±16. 25 | 111.03 ± 6.04 |
| | Diastolic BP (Mean ± SD) | 85.61 ± 9.28 | 73.71± 3.75 |
| Stage of blood pressure | Normal | 9 (8%) | 102 (100%) |
| | Prehypertension | 39(38.2%) | |
| | Stage 1 | 38(37.3%) | |
| | Stage 2 | 16(15.7%) | |
| Duration of illness | < 5 years | 52(51.0%) | |
| | 5–10 years | 35(34.3%) | |
| | >10 | 15(14.7%) | |

**Key**: Illiterate[d] = read and write only.

Finally, we compared hematological parameters between male control and hypertensive groups, and parameters such as WBC count, MCV and platelet count were normally

**Table 2. Comparison of hematological parameters between hypertensive and apparently healthy control group at HFSUH and Jugel hospital, Harar, eastern Ethiopia, 2020 (n = 204).**

| Parameters | Hypertensive (n = 102) | Healthy controls(n = 102) | P-value |
|---|---|---|---|
| | Median ± IQR | Median ± IQR | |
| WBC | 6.52 ± 3.08 | 5.29 ± 2.27 | 0.000 |
| RBC | 4.78± 0.79 | 4.70 ± 0.75 | 0.055 |
| HCT | 42.45 ± 5.42 | 40.60 ± 4.33 | 0.001 |
| HGB | 14.50 ± 1.93 | 13.78 ± 2.13 | 0.027 |
| MCV | 88.05± 8.84 | 88.55 ± 11.98 | 0.394 |
| MCH | 29.80 ± 2.78 | 30.40 ± 2.03 | 0.118 |
| MCHC | 34.00 ± 1.92 | 34.35 ± 3.95 | 0.192 |
| RDW | 13.85 ±1.60 | 13.60 ± 1.55 | 0.018 |
| RDW SD | 43.90 ± 5.30 | 42.03 ± 2.85 | 0.063 |
| MPV | 9.50 ± 2.22 | 9.04 ± 1.06 | 0.024 |
| | Mean ± SD | Mean ± SD | |
| PLT Count | 250. 47 ± 75.72 | 244.99 ± 82.24 | 0.714 |

**Table 3. Correlation of hematological parameters with blood pressure indices among hypertensive individuals at HFSUH and Jugel hospitals, Harar, Eastern Ethiopia, 2020 (n = 204).**

| Variables | Systolic blood pressure | Diastolic blood pressure |
|---|---|---|
| | Correlation coefficient (P-value) | Correlation coefficient (P-value) |
| WBC | .180 (0.070) | .102 (0.310) |
| RBC | .255 (0.010) ** | .241(0.015) * |
| HGB | .156 (0.117) | .183(0.065) |
| HCT | .219 (0.027) * | .354 (0.000) ** |
| MCV | -.162 (0.103) | .030 (0.767) |
| MCH | .076 (0.450) | .127 (.204) |
| MCHC | .145(0.147) | .181(0.069) |
| RDW | .418 (0.000) ** | .281(0.004) ** |
| RDW–SD | .186(0.061) | .152 (0.127) |
| PLT COUNT | —.070 (0.484) | .079 (0.430) |
| MPV | .046 (0.647) | .098 (0.329) |

**Key**: * = for p-value< 0.05,

** = for p-value < 0.01.

distributed while the rest of the parameters were not. The finding showed that parameters such as WBC count, MCV, MCH and MCHC were significantly different between the two groups (Table 5).

In case of females, only platelet count and RDW CV found to be normally distributed. Significant difference in WBC count, HCT and Hgb were observed bewteen female control and hypertensive group (Table 6).

## Discussion

Hypertension has become the major public health problem in developing countries [12]. In Ethiopia, hypertension related CVD death is increasing [13]. Thus, the need for early identification of patient with risk of cardiovascular disease and timely intervention is vital. So, this study

**Table 4. Correlation of hematological parameters with body mass index and duration of illness among hypertensive individuals at HFSUH and Jugel hospitals, Harar, eastern Ethiopia, 2020 (n = 204).**

| Variables | BMI r (P-value) | Duration of illness r (P-value) |
|---|---|---|
| WBC | .208(0.036) * | .155(0.119) |
| RBC | .114(0.253) | .126(0.206) |
| HGB | .034(0.736) | .125(0.211) |
| HCT | .011(0.915) | .060(0.552) |
| MCV | -.154(0.122) | -.064(0.524) |
| MCH | .014(0.890) | -.123(0.217) |
| MCHC | .019(0.850) | -.127(0.205) |
| RDW | .198(0.046) * | .049(0.622) |
| RDW–SD | .030(0.763) | -.075(0.455) |
| PLT COUNT | .190(0.056) | -.219(0.027) * |
| MPV | -.022(0.828) | .255(0.010) * |

**Key**: * = for p-value< 0.05,

** = for p-value < 0.01, r = correlation coefficient.

**Table 5. Comparison of hematological parameters between male hypertensive and male control group at HFSUH and Jugel hospital, Harar, Ethiopia, 2020 (n = 78).**

| Parameters | Hypertensive males (n = 39) | Healthy control males (n = 39) | P value |
|---|---|---|---|
| | Mean ± SD | Mean ± SD | |
| WBC (10³/ul) | 6.82± 2.02 | 5.30 ± 1.50 | 0.013 |
| PLT Count(10³/ul) | 241.95 ± 67.01 | 234.25 ± 66.37 | 0.972 |
| MCV (fl) | 89.37± 4.48 | 86.81± 7.64 | 0.030 |
| MCH (pg) | 30.48± 1.40 | 31.10± 2.70 | 0.040 |
| | Median ± IQR | Median ± IQR | |
| HCT (%) | 45.10± 7.20 | 41.80 ± 6.00 | 0.029 |
| Hgb (g/dl) | 15.50 ± 2.60 | 15.03 ± 2.33 | 0.404 |
| MCHC(g/dl) | 34.7± 1.60 | 36.2 ± 1.40 | 0.001 |
| RDW CV (%) | 13.80 ±1.40 | 14.00± 0.90 | 0.213 |
| RDW SD (fl) | 43.20 ± 5.00 | 42.60 ± 6.30 | 0.700 |
| MPV (fl) | 9.50 ± 2.30 | 8.80 ± 1.32 | 0.072 |
| RBC (10⁶/ul) | 5.20 ± 0.86 | 4.89 ± 0.65 | 0.146 |

is aimed to compare hematological profile of hypertension patients with apparently healthy individuals as well as to correlate blood pressure indices with hematological parameters.

In this study, the median values of WBC count, Hgb, HCT,RDW and MPV were significantly higher in hypertensive patients compared to apparently healthy control group. In the bivariate correlation analysis, RBC count, HCT and RDW showed positive correlation with blood pressure indices. RDW and WBC count also achieved significant positive correlation with BMI. On the other hand, duration of hypertension was positively and negatively correlated with MPV and PLT count, respectively.

Our finding showed that hypertensive group had significantly higher median ± IQR value of WBC count (6.52 ± 3.08) compared to the control group (5.29 ±2.27). The finding is in agreement with Enawegaw et al. [18], Babu et al. [15], Al muhana et al. [19] and Emamian et al. [7]. In contrast to this, another studies by Reis RS et al. [20] and Divya R et al. [21] showed no statistically significant result in WBC count. The increased WBC count in hypertensive patient could be due to the presence of vascular dysfunction in those patients which leads to

**Table 6. Comparison of hematological parameters between female hypertensive and female control group at HFSUH and Jugel hospital, Harar, Ethiopia, 2020 (n = 78).**

| Parameters | Hypertensive females (n = 63) | Healthy control females (n = 63) | P-value |
|---|---|---|---|
| | Median ± IQR | Median ± IQR | |
| WBC (10³/ul) | 6.40 ± 2.62 | 5.17± 1.97 | 0.001 |
| RBC count | 4.69 ±0.53 | 4.58 ±0.70 | 0.104 |
| MCV (fl) | 88.10 ±10.70 | 89.70± 10.80 | 0.073 |
| MCH (pg) | 29.65 ± 3.10 | 30.30 ± 2.00 | 0.150 |
| HCT (%) | 42.00 ± 4.90 | 40.00 ± 4.00 | 0.010 |
| Hgb (g/dl) | 14.01 ± 1.60 | 13.20 ± 1.30 | 0.012 |
| MCHC(g/dl) | 33.40± 1.50 | 33.30 ± 3.10 | 0.414 |
| RDW SD (fl) | 44.30 ± 5.70 | 42.03 ± 1.30 | 0.044 |
| MPV (fl) | 9.50 ± 1.35 | 9.05 ±0.71 | 0.108 |
| | Mean ± SD | Mean ± SD | |
| PLT Count(10³/ul) | 255.75 ± 89.71 | 251.63 ± 90.56 | 0.439 |
| RDW CV (%) | 14.13 ± 1.27 | 13.20 ±1.31 | 0.928 |

activation of cytokine system. cytokines such as SCF/c-kit are produced to repair the endothelial injury; participate in differentiation and proliferation of hematopoietic cell [22]. Activated and differentiated leucocytes can produce more cytokines and the increased adherence of the stimulated leukocytes to the vascular endothelium; causing capillary leukocytosis and subsequent increased vascular resistance, thus, causing an increase in blood pressure [23]. Activated WBCs also reflect the inflammatory activity of atherosclerosis that perpetuates vascular injury and tissue ischemia [24].

In the present study, the median± IQR values of Hgb were found to be significantly higher in the hypertensive (14.50±1.93) than in control (13.78±2.13) group. Studies from India [15], Iran [7], Korea [9] and Ethiopia [8] found similar results. In contrast, studies from Brazil [20], Saudi [19] and India [21] Showed contradicting result. The reason behind the increment of Hgb in hypertensive patients are not entirely known; but it might be related to endothelial cell damage and subsequent increased in the concentrations of growth factors [25]. Evidences show that concentration of serum hepatocyte growth factor is positively associated with hypertension and increased Hgb concentration. As growth factors enhance hematopoiesis, which produces erythrocytes, hemoglobin levels may increase with increasing levels of growth factors [26, 27].

In this study, the median± IQR values of HCT were found to be significantly higher in the hypertensive (42.45±5.42) than control (40.60±4.33) group. Previous studies also showed similar results [7, 8, 15, 28]. On the other hand, study by Divya et al. [21] found significantly lower HCT levels in hypertensive patients compared with control group. The possible mechanisms underlying the association between HCT and high blood pressure is that, HCT is a determinant factor for high whole blood viscosity during hypertension [29]. This may lead to a peripheral resistance to blood flow and high blood pressure [30]. Evidences showed that, most hypertensive patients exhibit increased blood viscosity compared with healthy controls. Therefore, high HCT in hypertension could reflect a true increase in red blood cell mass as well as hemoconcentration caused by a reduction in plasma volume [28].

In our study, median ± IQR of RDW increased significantly in hypertensive group (13.90 ± 1.65) than in a healthy control (13.60 ± 1.55) group. This result is in line with studies by Tanindi A. et al. [31] and Gunebakmaz O.et al. [32]. Another study from Iran by Emanain M. et al. [7] reported contradicting result. The increased RDW is supported by evidences that suggest higher RDW result from ineffective erythropoiesis due to chronic inflammation [33]. Inflammatory cytokines have been found to smother the maturation of erythrocytes, which enable juvenile red cells to enter into the circulation and increases the heterogeneity in size [34]. In addition, raised RDW may reflect enhanced erythropoiesis coming about because of the circulating levels of neurohormonal mediators, which lead to an increase in the heterogeneity of circulating red cells [35].

This study also revealed that the median± IQR value of MPV is higher in hypertensive patients (9.55 ± 2.22) compared to control group (9.04 ± 1.06). This result is in line with a study conducted in Gondar, Ethiopia [8]. The possible mechanisms for the increased MPV could relate to vascular injury in hypertensive patients. High blood pressure causes endothelial damage, which leads to increased PLT activation and initiation of PLT production. Evidences suggest that PLT consumption increases at the site of injured blood vessel; which causes larger PLTs to escape from the bone marrow. This leads to increment in PLT count and MPV. Since larger PLTs are hemostatically more active than mature one, their presence becomes a risk factor for development of coronary thrombosis and myocardial infraction [36, 37].

As to the correlation of hematological parameters and blood pressure indices, HCT level was significantly and positively correlated with both systolic (r = 0.219, P = 0.027) and diastolic (r = 0.354, P = 0.00) blood pressures. This finding is in line with studies from Ethiopia [8] and Italy [28]. A study from Nigeria [38] however, found a negative correlation between HCT and

blood pressure indices. The exact reason behind positive correlation of HCT with blood pressure is still unknown but it might be related to an increase in red blood cell mass which increases with blood pressure.

RBC count was also weakly and positively correlated with systolic (r = 0.255, P = 0.01) and diastolic (r = 0.241, P = 0.015) blood pressures in our study. This result is in accordance with Enawegaw et al. [8].This association could be explained by the augmentation of growth factors due to distended vascular injury caused by increased blood pressure [6]. RDW was also positively correlated with systolic (r = 0.418, P = 0.000) and diastolic (r = 0.281, P = 0.004) blood pressure. This positive correlation could be related to the amplified chronic inflammation which comes from increased blood pressure [33].

With respect to the correlation of hematological indices with BMI in hypertensive patients, WBC count achieved positive correlation with BMI (r = 0.208). This could be due to the fact that adipose tissue is a great source of inflammatory factors, such as interleukin(IL)-6 and IL-8, which are also important inducers of WBC production [39]. RDW also achieved significant positive correlation with BMI (r = 0.198). This might be due to the fact that, obesity is characterized by chronic, low grade, systemic inflammation which could lead to impaired erythropoiesis there by elevating RDW [40].

In present study, duration of illness since diagnosis of hypertension achieved significant but negative correlation with platelet count (r = -0.219, P = 0.027) and positive correlation with MPV (r = 0.255, P = 0.01). This could be due to the fact that, vascular complication in hypertensives worsens with longer duration. This means, the vascular injury and consequent platelet activation leads to consumption of platelets and escaping of larger PLTs from the bone marrow [41]. This could lead to lower platelet count and high MPV.

Finally, the effect of hypertensive treatment on the hematological parameters were not assessed and using this study as baseline, more advanced studies that include lipid profile of patient and control are required to address these problems and to fully comprehend the role of hematological parameters in the prognosis and management of hypertension related complication.

## Conclusion

In the present study, the median values of WBC count, Hgb, HCT t, RDW and MPV was significantly higher in hypertensive patient compared with healthy controls. In the bivariate spearman correlation analysis, RBC count, HCT and RDW showed significant correlation with blood pressure indices. Duration of illness also achieved significant correlation with platelet count (r = - 0.219) and MPV (r = 0.255). Thus, investigation of hematological parameters might help in earlier identification and timely intervention of hypertension complication related to abnormal hematological parameters. However, further studies are required to understand hypertensive associated changes in hematological parameters.

## Supporting information

**S1 Annex. Questionnaire for hypertension patients and controls.**
(DOCX)

**S2 Annex. SPSS data sheet.**
(SAV)

## Acknowledgments

We would also like to extend our gratitude to the hospital heads, the study participants and the data collectors for their collaboration.

## Author Contributions

**Conceptualization:** Beza Sileshi, Fekadu Urgessa, Moges Wordofa.

**Data curation:** Beza Sileshi.

**Formal analysis:** Beza Sileshi.

**Funding acquisition:** Beza Sileshi.

**Investigation:** Beza Sileshi, Fekadu Urgessa, Moges Wordofa.

**Project administration:** Beza Sileshi, Fekadu Urgessa, Moges Wordofa.

**Software:** Beza Sileshi.

**Supervision:** Fekadu Urgessa, Moges Wordofa.

**Validation:** Beza Sileshi, Fekadu Urgessa, Moges Wordofa.

**Writing – original draft:** Beza Sileshi.

**Writing – review & editing:** Beza Sileshi, Fekadu Urgessa, Moges Wordofa.

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
