## [Decision Letter · Decision Letter 0]

7 Apr 2021

PONE-D-21-02215

A comparative study of hematological parameters between hypertensive and normotensive individuals in Harar, eastern Ethiopia.

PLOS ONE

Dear Dr. Gelan,

Your manuscript has now been reviewed by two referees who have raised concerns over the lack of detail regarding the sampling technique, as well as the definitions for hypertensives and control patients. There is also some ambiguity over the number of males and females in this study. Since this is fairly important information I invite you to submit a revised version of the manuscript that addresses the points raised by the reviewers and includes the needed information.

We look forward to receiving your revised manuscript.

Kind regards,

Colin Johnson, Ph.D.

Academic Editor

PLOS ONE

Journal Requirements:

http://citeseerx.ist.psu.edu/viewdoc/download?doi=10.1.1.960.2896&rep=rep1&type=pdf

https://link.springer.com/article/10.1186/s12878-017-0093-9

https://www.wjgnet.com/2220-3168/full/v5/i2/93.htm

In your revision ensure you cite all your sources (including your own works), and quote or rephrase any duplicated text outside the methods section.

Further consideration is dependent on these concerns being addressed.

4. Please provide a sample size and power calculation in the Methods, or discuss the reasons for not performing one before study initiation.

5. Please include additional information regarding the survey or questionnaire used in the study and ensure that you have provided sufficient details that others could replicate the analyses. For instance, if you developed a questionnaire as part of this study and it is not under a copyright more restrictive than CC-BY, please include a copy, in both the original language and English, as Supporting Information.

6. We note that you have indicated that data from this study are available upon request. PLOS only allows data to be available upon request if there are legal or ethical restrictions on sharing data publicly. For information on unacceptable data access restrictions, please see http://journals.plos.org/plosone/s/data-availability#loc-unacceptable-data-access-restrictions.

7. Your ethics statement should only appear in the Methods section of your manuscript. If your ethics statement is written in any section besides the Methods, please delete it from any other section.

Reviewers' comments:

Reviewer's Responses to Questions

**Comments to the Author**

1. Is the manuscript technically sound, and do the data support the conclusions?

Reviewer #1: Yes

Reviewer #2: Partly

2. Has the statistical analysis been performed appropriately and rigorously? 

Reviewer #1: Yes

Reviewer #2: I Don't Know

3. Have the authors made all data underlying the findings in their manuscript fully available?

Reviewer #1: Yes

Reviewer #2: Yes

4. Is the manuscript presented in an intelligible fashion and written in standard English?

Reviewer #1: Yes

Reviewer #2: No

5. Review Comments to the Author

Reviewer #1: 1. Authors did not show how they calculated the sample size. They should explain how they got the sample size.

2. Authors did not state the number of males and female in both cases and controls. This should be stated in the manuscript.

3. The sampling techniques was not well described in the methodology. The mention of convenient sampling is not enough.

4. The criteria for selecting the controls is not clearly described in the methodology section. Authors must describe this.

5. The operational definitions for the cases (hypertensives) and controls (normotensive) are not stated clearly stated. Authors must describe this.

Reviewer #2: I have read your manuscript with interest. In general, the paper addresses an issue that admittedly needs more study and attention. This is a manuscript where the authors examined the "A comparative study of hematological parameters between hypertensive and normotensive individuals in Harar, eastern Ethiopia".

1. While appreciating the challenge of preparing a manuscript in the English language for many authors, the paper should be carefully edited by a native English-speaking editor.

2. No information about the physical activity records.

3. Please explain more about the implication of this study results. How can the study results help clinically?

4. The Discussion would be improved by detailing more specific implications for future studies.

5. The limitation of the study should be fully explained in the discussion section.

6. Please describe more about the novelty of manuscript in the introduction. It is very important for readers to know about this novelty.

6. PLOS authors have the option to publish the peer review history of their article (what does this mean?). If published, this will include your full peer review and any attached files.

Reviewer #1: No

Reviewer #2: No

---

## [Author Response · Author response to Decision Letter 0]

29 Jun 2021

Cover letter

Moges Wordofa

Addis Ababa University

Addis Ababa, Ethiopia 

Email: heranmakmow@gamil.com or moges.wordofa@aau.edu.et

Date: May 21, 2021

To: PLOS ONE 

Dear Editorial: 

We are glad to write this response to our paper entitled as “A comparative study of hematological parameters between hypertensive and normotensive individuals in Harar, eastern Ethiopia” (Manuscript ID: PONE-D-21-02215) which has been requested to review for publication in PLOS ONE journal. We are pleased to have an opportunity to make our paper revised and we have greatly appreciated the editor’s comments and suggestions which were considered as very helpful. In revising the paper, we have carefully considered your comments and suggestions on our revised submission.

 Response to editorial correction

We have gone through the manuscript to edit and amend some editorial correction. By doing so, we have corrected language error and other topographical errors in the document. We have also prepared the manuscript according to PLOS ONE style requirements. Besides, if there is any error that we missed, we are very ready to correct it again. 

Response to the reviewer’s comment

Dear Reviewers: 

We are glad to write this response to our paper entitled as “A comparative study of hematological parameters between hypertensive and normotensive individuals in Harar, eastern Ethiopia” (Manuscript ID: PONE-D-21-02215) which has been requested to review for publication in PLOS ONE journal. We are pleased to have an opportunity to make our paper revised and we have greatly appreciated your comments and suggestions which were considered as very helpful. In revising the paper, we have carefully considered the comments and suggestions on our revised submission. As instructed, we have attempted to thoroughly explain changes made to all comments and reply to each comment in point-by-point fashion as follows: 

Response to Reviewer#1

Comment#1. “Authors did not show how they calculated the sample size. They should explain how they got the sample size.”

Response#1: The sample size was determined from a study in Indian based on mean and variance of hemoglobin. We used 95% CI and power of 0.8 as well as based on formula for double population of equal size mean. However, we enrolled additional participants other than the one we obtained using the formula. The purpose was to increase presentiveness of the sample to the general population. 

Comment #2. “Authors did not state the number of males and female in both cases and controls. This should be stated in the manuscript.”

Response #2: The total number of males and females in both normotensive and hypertensive group were equal and it was 39 and 63, respectively. we have already included this in the result section of the manuscript in a previous submission.

Comment #3. “The sampling techniques was not well described in the methodology. The mention of convenient sampling is not enough.”

Response #3. As suggested by the reviewer, we have elaborated the technique in the methodology section.

Comment#4. “The criteria for selecting the controls are not clearly described in the methodology section. Authors must describe this” 

Response#4. As suggested by the reviewer, we have clarified the donor selection criteria in methodology part of the manuscript. 

Comment#5. “The operational definitions for the cases (hypertensives) and controls (normotensive) are not stated clearly stated. Authors must describe this”

Response#5. As suggested by the reviewer, we have included the definition of normotensive and hypertensive under operational definition in methodology section

Response to Reviewer#2

Comment#1: “While appreciating the challenge of preparing a manuscript in the English language for many authors, the paper should be carefully edited by a native English-speaking editor”

Response#1: The manuscript is thoroughly revised by native English writer in the field and linguistic errors are corrected

Comments#2: “No information about the physical activity records”.

Response#2: we thought it is irrelevant to the study

Comment#3: “Please explain more about the implication of this study results. How can the study results help clinically?”

Response#3: As suggested by the reviewer, the significance of the study is addressed in the introduction part of the manuscript.

Comment#4: “The Discussion would be improved by detailing more specific implications for future studies”

Response#4: It has been included as it could be used as a baseline for performing further advanced studies including lipid profile and others

Comment#5: “The limitation of the study should be fully explained in the discussion section”

Response#5: We have included the limitation of the study in the discussion 

Comment#6: “Please describe more about the novelty of manuscript in the introduction. It is very important for readers to know about this novelty.”

Response#6: We have tried to include novelty and necessity of this study among hypertensive patients in the country and worldwide. 

Looking forward to hearing from you. Thank you again for your consideration! 

Sincerely, 

Moges Wordofa

---

## [Decision Letter · Decision Letter 1]

27 Jul 2021

PONE-D-21-02215R1

A comparative study of hematological parameters between hypertensive and normotensive individuals in Harar, eastern Ethiopia .

PLOS ONE

Dear Dr. Gelan,

Your revised manuscript has been reviewed, and neither reviewer recommended publication of the revision. Specifically concern has been raised regarding separation of males and females during analysis.  If you believe you can address this issue, submit a revised version of the manuscript that addresses the point raised.

We look forward to receiving your revised manuscript.

Kind regards,

Colin Johnson, Ph.D.

Academic Editor

PLOS ONE

Reviewers' comments:

Reviewer's Responses to Questions

**Comments to the Author**

1. If the authors have adequately addressed your comments raised in a previous round of review and you feel that this manuscript is now acceptable for publication, you may indicate that here to bypass the “Comments to the Author” section, enter your conflict of interest statement in the “Confidential to Editor” section, and submit your "Accept" recommendation.

Reviewer #1: All comments have been addressed

Reviewer #2: (No Response)

2. Is the manuscript technically sound, and do the data support the conclusions?

Reviewer #1: Partly

Reviewer #2: (No Response)

3. Has the statistical analysis been performed appropriately and rigorously? 

Reviewer #1: Yes

Reviewer #2: (No Response)

4. Have the authors made all data underlying the findings in their manuscript fully available?

Reviewer #1: Yes

Reviewer #2: (No Response)

5. Is the manuscript presented in an intelligible fashion and written in standard English?

Reviewer #1: Yes

Reviewer #2: (No Response)

6. Review Comments to the Author

Reviewer #1: my comments is that when dealing with hematological parameters, authors will have to seperate male from females in the analysis. hematological parameters for males and female cannot be analysed together they must be separated.

Reviewer #2: (No Response)

7. PLOS authors have the option to publish the peer review history of their article (what does this mean?). If published, this will include your full peer review and any attached files.

Reviewer #1: No

Reviewer #2: No

---

## [Author Response · Author response to Decision Letter 1]

14 Oct 2021

I have tried to address the comment suggested by the reviewer and I have included the correction in the revised manuscript

Thank you

---

## [Editor Report · Decision Letter 2]

17 Nov 2021

A comparative study of hematological parameters between hypertensive and normotensive individuals in Harar, eastern Ethiopia .

PONE-D-21-02215R2

Dear Dr. Gelan,

We’re pleased to inform you that your manuscript has been judged scientifically suitable for publication and will be formally accepted for publication once it meets all outstanding technical requirements.

Kind regards,

Colin Johnson, Ph.D.

Academic Editor

PLOS ONE
---

## [Editor Report · Acceptance letter]

22 Nov 2021

PONE-D-21-02215R2 

A comparative study of hematological parameters between hypertensive and normotensive individuals in Harar, eastern Ethiopia. 

Dear Dr. Wordofa:

I'm pleased to inform you that your manuscript has been deemed suitable for publication in PLOS ONE. Congratulations! Your manuscript is now with our production department. 

Kind regards, 

on behalf of

Dr. Colin Johnson 

Academic Editor

PLOS ONE